# Critical Assessment of Cancer Characterization and Margin Evaluation Techniques in Brain Malignancies: From Fast Biopsy to Intraoperative Flow Cytometry

**DOI:** 10.3390/cancers15194843

**Published:** 2023-10-03

**Authors:** Ioannis Liaropoulos, Alexandros Liaropoulos, Konstantinos Liaropoulos

**Affiliations:** 1Ippokrateio General Hospital of Athens, 115 27 Athens, Greece; jliaropoulosabc@gmail.com; 2Neurosurgical Department, Athens Euroclinic, 115 21 Athens, Greece; aliaropoulosabc@gmail.com

**Keywords:** fast biopsy, margin evaluation, 5-aminovulenic acid, intraoperative magnetic resonance imaging, intraoperative flow cytometry

## Abstract

**Simple Summary:**

This review delves into novel techniques in brain malignancy management. Fast biopsy techniques offer a rapid and minimally invasive diagnosis, while intraoperative flow cytometry provides real-time cellular insights during surgeries. Intraoperative MRI integrates real-time imaging into surgical procedures, and 5-aminolevulinic acid enhances tumor visualization through fluorescence. Each method presents unique advantages, from speed and precision to enhanced visualization, but also comes with inherent challenges. As technology advances, these techniques are poised for further refinement, promising improved patient outcomes. Collaborative research will be instrumental in driving future innovations in brain tumor diagnosis and treatment.

**Abstract:**

Brain malignancies, given their intricate nature and location, present significant challenges in both diagnosis and treatment. This review critically assesses a range of diagnostic and surgical techniques that have emerged as transformative tools in brain malignancy management. Fast biopsy techniques, prioritizing rapid and minimally invasive tissue sampling, have revolutionized initial diagnostic stages. Intraoperative flow cytometry (iFC) offers real-time cellular analysis during surgeries, ensuring optimal tumor resection. The advent of intraoperative MRI (iMRI) has seamlessly integrated imaging into surgical procedures, providing dynamic feedback and preserving critical brain structures. Additionally, 5-aminolevulinic acid (5-ALA) has enhanced surgical precision by inducing fluorescence in tumor cells, aiding in their complete resection. Several other techniques have been developed in recent years, including intraoperative mass spectrometry methodologies. While each technique boasts unique strengths, they also present potential limitations. As technology and research continue to evolve, these methods are set to undergo further refinement. Collaborative global efforts will be pivotal in driving these advancements, promising a future of improved patient outcomes in brain malignancy management.

## 1. Introduction

Brain malignancies (BMs) are a diverse group of neoplasms originating either from brain tissue and/or its surrounding structures or metastasizing from other parts of the body [1]. BMs represent a significant challenge in the field of oncology. Their intricate location, coupled with the delicate nature of the brain tissue, necessitates precise and accurate diagnostic and therapeutic approaches. Because these tumors can be primary or secondary and their diagnosis and treatment depend on various factors, including tumor location, size, and type, the prognosis and treatment outcomes for patients with brain tumors largely depend on the early and accurate characterization of the malignancy and the precise evaluation of tumor margins during surgical interventions [2,3,4,5].

Brain malignancies are a significant health concern, with primary brain tumors accounting for more than 300,000 new cases and more than 250,000 deaths in the year 2020, according to Global Cancer Observatory [6]. The diagnosis and treatment of brain malignancies present unique challenges due to the sensitive and complex nature of the brain [7,8]. On the one hand, accurate characterization of the malignancy would dictate following treatment steps, while on the other, it is also critical to accurately characterize and evaluate the margins of each BM to minimize the risk of recurrence [9,10].

The evolution of diagnostic techniques has seen a shift from traditional methods to more advanced procedures. Novel fast biopsy techniques have emerged as a promising alternative, offering rapid diagnosis with minimized patient discomfort. Another promising advancement includes 5-aminovulenic acid (5′ALA) analysis as well as the intraoperative magnetic resonance imaging (iMRI) technique. On the other hand, intraoperative flow cytometry (iFC), a relatively novel approach, provides near-real-time data during surgical procedures, aiding in the immediate assessment of tumor margins and ensuring maximum tumor resection with minimal damage to healthy tissues. From stereotactic biopsy to surgical management, novel tumor evaluation techniques and technologies hold a promise to further evolve the field for the benefit of the patient [11,12,13,14,15,16].

This review aims to critically assess such pivotal next-generation tumor characterization and margin evaluation techniques, shedding light on their development, advantages, limitations, and clinical applications. Furthermore, by comparing fast intraoperative techniques, we endeavor to provide a comprehensive understanding of their roles in the current landscape of brain malignancy management and their potential future routes.

## 2. Fast Biopsy Techniques

### 2.1. Background

Fast biopsy techniques, as the name suggests, prioritize speed without compromising the accuracy of tissue sampling. Originating in the late 20th century, these techniques were developed in response to the need for rapid diagnostic procedures that could provide immediate insights into the nature of brain tumors. Traditional biopsy methods, while thorough, often required extended waiting periods for histopathological analysis, delaying potential treatment interventions A type of fast biopsy, also known as stereotactic biopsy or needle biopsy, is a minimally invasive technique used to obtain tissue samples from brain tumors. This technique can provide a diagnosis with high accuracy, up to 98%, but the technique has limitations in terms of tumor characterization and margin evaluation. Fast biopsy provides a small tissue sample that may not represent the entire tumor, which can lead to sampling error. Additionally, the biopsy does not provide information on the tumor’s margin or invasion into surrounding brain tissue, which can impact treatment decisions. Fast biopsy is also limited in the ability to differentiate between different tumor grades and types, as this requires a larger tissue sample and a detailed pathology assessment [11,12,17].

Fast biopsy assessment can also be intraoperative, by rapid immunocytochemistry [18]. The evolution of fast biopsy techniques was driven by advancements in imaging modalities, molecular biology, and tissue processing methods. These techniques, often guided by real-time imaging, such as MRI or CT scans, allow for targeted sampling of tumor tissues, ensuring that representative samples are obtained with minimal invasiveness [19,20,21]. However, it is essential to properly prepare for and manage any potential brain tumor sample during intraoperative consultation. This not only offers vital diagnostic details that can guide the neurosurgeon’s decisions during the procedure but also ensures that any additional studies needed for diagnosis and patient care can be effectively conducted [21].

### 2.2. Advantages, Limitations, and Clinical Applications

The primary advantage of fast biopsy techniques lies in their rapidity. Clinicians can obtain diagnostic insights within hours, if not minutes, facilitating timely therapeutic decisions. This speed is particularly crucial in cases where immediate surgical or therapeutic interventions are necessary based on the biopsy results, such as in higher-grade tumors. Moreover, the minimally invasive nature of stereotactic biopsy, along with the minimal tissue needed for intraoperative fast biopsy reduces the risk of complications, such as infections or hemorrhages, and ensures quicker patient recovery. The integration of novel modalities, such as real-time imaging, further enhances the accuracy of tissue sampling, minimizing the chances of nonrepresentative sampling. However, fast biopsy techniques are not without limitations. The rapid processing of tissues might sometimes compromise the quality of histopathological slides, potentially leading to diagnostic inaccuracies. Additionally, while these techniques are less invasive than traditional methods, they still carry inherent risks associated with any invasive procedure [17,21]. In the clinical setting, fast biopsy techniques have revolutionized the approach to brain tumor diagnosis. They are especially valuable in emergency scenarios where rapid diagnosis can be the difference between life and death. For instance, in cases of suspected high-grade gliomas or rapidly progressing tumors, a fast biopsy can provide immediate clarity, guiding surgeons on the best follow-up [22]. Furthermore, these techniques have found applications in monitoring the progression or regression of known tumors, especially when patients present with new or worsening neurological symptoms. By providing a quick snapshot of the tumor’s current state, clinicians can adjust treatment plans accordingly, ensuring optimal patient outcomes.

## 3. Emerging Techniques in Brain Tumor Surgery

### 3.1. Image-Guided Surgery and 5′ALA

Image-guided surgery, also known as neuronavigation, is a technique that uses advanced imaging technology to assist surgeons in navigating the brain and locating tumors during surgery [23]. This technique allows for real-time visualization of the tumor and surrounding brain tissue, enabling surgeons to more accurately identify tumor margins and areas of infiltration. Image-guided surgery can improve the accuracy of surgical resection and minimize damage to healthy brain tissue. However, this technique also has limitations, particularly in the identification of tumor margins. The imaging technology used for image-guided surgery may not be able to detect small tumor infiltrations or microscopic tumor cells, which can lead to incomplete resection and a higher risk of recurrence [24,25].

To this end, fluorescence agents may assist toward accuracy of surgical treatment. 5-aminolevulinic acid, or 5-ALA, is a photosensitizing agent that has gained traction in fluorescence-guided brain tumor surgeries. When administered, it causes tumor cells to produce a fluorescent compound, making them glow under specific light wavelengths, thus providing a means to detect cancer and to delineate tumor margins. This fluorescence provides a contrast between malignant and healthy tissues [14]. The use of 5-ALA enhances the surgeon’s ability to visualize and differentiate tumor margins, leading to more complete resections. This is especially beneficial in cases of aggressive tumors, where clear delineation between healthy and malignant tissues is crucial.

Meningiomas are the most prevalent primary tumors in the central nervous system. Achieving complete removal is curative, but spotting leftover tumor parts during surgery is challenging. Meningiomas, while often benign, can recur and result in poor outcomes due to their proximity to vital neurovascular structures, leading to incomplete removal and the need for additional treatments. 5′ALA has emerged as a tool to enhance the likelihood of full tumor removal while reducing unintended damage. A comprehensive review and meta-analysis of the literature were conducted by Foster et al., with 19 studies meeting the criteria and 222 patients in total. The findings showed that 5′ALA is both highly specific and sensitive (95%). It played a pivotal role in adjusting surgical plans, especially in high-grade meningiomas, and was effective in distinguishing between tumor growth and invasion of surrounding areas. In conclusion, 5′ALA significantly aids in meningioma surgeries [26]. Several other dyes are also under development. Lee et al. introduced a new method using a near-infrared (NIR) fluorescent dye called “second-window indocyanine green” (ICG) to locate tumor tissue. Patients received this dye a day before surgery, and an NIR camera was employed during the operation. Of the 18 patients studied, 14 showed a strong signal-to-background ratio (SBR) for the tumor compared to the surrounding brain. The technique had a 96.4% sensitivity for detecting the tumor. The results suggested that using this dye before surgery can enhance intraoperative visualization of meningiomas, potentially improving surgical outcomes [27].

Among brain cancers, glioblastoma multiforme (GBM) is a highly lethal human cancer. A meta-analysis of fluorescence image-guided surgical resection (FIGR) using 5-ALA showed promising results. From 20 selected studies, the gross total resection rate was 75.4%, and the time to tumor progression averaged 8.1 months. The technique demonstrated high specificity (88.9%) and sensitivity (82.6%), suggesting that 5-ALA-FIGR can significantly improve patient outcomes in GBM [28]. This level of accuracy is also consistent in individual studies, showing that 5′ALA improves image-guided surgery diagnostic potential [29,30].

However, the fluorescence produced by 5-ALA is not universally consistent across all tumor types. Some tumors might not produce sufficient fluorescence, and there is also the potential for false positives in certain scenarios. However, the promising results so far have helped 5-ALA to become a useful adjunct in many neurosurgical procedures, especially for high-grade gliomas. Its ability to highlight tumor cells has led to improved resection rates and, consequently, better post-operative outcomes for patients [30,31].

### 3.2. Intraoperative MRI (iMRI)

Intraoperative MRI, commonly referred to as iMRI, has emerged as a revolutionary tool in the field of neurosurgery over the past two decades. Designed to provide real-time imaging during surgical procedures, iMRI allows surgeons to visualize the tumor and surrounding brain structures with unparalleled clarity, even as surgical interventions are underway [13].

The primary advantage of iMRI lies in its ability to offer dynamic feedback. Surgeons can assess the extent of tumor resection in real-time, ensuring maximal removal while preserving critical brain structures. This real-time feedback can be pivotal in reducing the need for follow-up surgeries. In addition, resecting primary brain tumors near vital areas is challenging. A systematic review and meta-analysis study by Tuleascaa et al., which included a total of 527 patients, showed that combining awake craniotomy with intraoperative MRI resulted in a 56.3% complete radiological resection rate and highlighted its benefits for tumors near language regions. Despite technical challenges, experienced teams can effectively use this approach to optimize tumor removal while preserving neurological function during glioma resection [32]. In another study comparing intraoperative magnetic resonance imaging with conventional neuronavigation-guided resection for glioma patients, iMRI led to a higher rate of total tumor removal and improved 6-month progression-free survival (PFS). However, there was no significant difference in the extent of resection, tumor size reduction, surgery duration, or 12-month overall survival between the two methods. The study’s limitations included data from nonrandomized controlled trials and a small, diverse sample size [33].

However, integrating MRI technology into the surgical suite presents several challenges. The equipment needs space and is expensive, and the surgical team needs to be trained to operate within the magnetic field. Additionally, the cost implications of setting up and maintaining an iMRI-equipped operating room can be significant. In the clinical setting, iMRI has proven invaluable for surgeries involving tumors located near critical brain structures. By providing a real-time view of the surgical site, it aids surgeons in making informed decisions, ensuring optimal patient outcomes [13,34].

### 3.3. Other Methodologies

There are several other methodologies that hold promise to increase the accuracy of brain tumor resection, including Raman spectroscopy-based methods and the use of exoscopes [35,36,37].

An interesting technology that has been recently developed is the application of mass spectrometry in brain tumor surgery. A good paradigm of this technology is the iKnife, or “intelligent knife,” which is a groundbreaking surgical tool that has garnered attention in recent years. Developed as an evolution of the traditional electrosurgical knife, the iKnife is equipped with a mass spectrometer that can analyze the smoke produced during electrosurgery. This real-time analysis allows the iKnife to differentiate between healthy and malignant tissues based on their distinct metabolic profiles [38].

The primary benefit of the iKnife lies in its ability to provide immediate feedback during surgeries. Surgeons can ascertain, in real-time, whether the tissue being cut is healthy or cancerous. This immediate differentiation can be crucial in ensuring complete tumor resection while minimizing damage to surrounding healthy tissues. Another interesting aspect is the use of instant learning approaches to improve accuracy [39]. While the iKnife offers a promising approach to tumor surgeries, it is not without challenges. The technique requires specialized equipment and training. Additionally, the metabolic profiles used for differentiation might vary across patients and tumor types, necessitating continuous calibration and refinement of the tool [40]. In the clinical setting, the iKnife has shown potential in surgeries involving various types of tumors, not just those in the brain. Its ability to provide real-time tissue differentiation makes it a valuable tool, especially in surgeries where tumor margins are challenging to delineate [41].

## 4. Intraoperative Flow Cytometry

Flow cytometry can quantify several cellular features [42,43], including measurable residual disease, lymph node and metastatic progression of cancer, circulating tumor cells, and DNA content and ploidy [44,45]. The use of flow cytometry technologies requires a flow cytometer, a number of specific reagents for each type of analysis, such as fluorochromes, and powerful software for data integration and analysis [42,46,47].

Based on the speed and accuracy of flow cytometry, a recent advancement is its use as an intraoperative adjunct in the diagnosis of several types of malignancies. Among the first applications of intraoperative flow cytometry (iFC) is the analysis of brain tumors [15,48]. iFC represents a fusion of surgical precision to identify the features for advanced cellular analysis. Introduced in the early 21st century, it has emerged as a groundbreaking technique in the realm of brain tumor surgeries. Unlike traditional histopathological methods that require tissue samples to be sent to a lab for analysis, iFC allows for near-real-time cellular evaluation directly within the operating room [15]. Apart from brain malignancy analysis [49], which includes the grade of meningiomas, a low malignancy in the brain [50], and gliomas, a mostly malignant and aggressive primary brain cancer [51], iFC has been successfully used in other malignancies, including head and neck tumors, breast cancer, gynecological malignancies, bladder cancer, and liver cancer and pancreatic, gastric, and colorectal cancer [51,52,53,54,55,56,57,58,59]. The principle behind iFC is the use of flow cytometry to analyze cells from the surgical site, distinguishing between malignant and nonmalignant cells based on DNA content features or other markers. This immediate feedback assists surgeons in determining tumor margins and making informed decisions about the extent of the resection required.

The most significant advantage of iFC in neurosurgery is its ability to provide near-real-time feedback during surgery. This immediacy can be crucial in ensuring maximal tumor resection while preserving healthy brain tissue, thereby improving post-operative outcomes and reducing recurrence rates. As regards meningiomas, a recent study by Alexiou et al. explored the use of intraoperative flow cytometry, specifically the ‘Ioannina Protocol’, to distinguish between low-grade (grade 1) and high-grade (grades 2–3) meningiomas in 59 patients. The results, available within 5 min, showed that high-grade meningiomas had distinct cell cycle phase differences compared to low-grade ones. Using a tumor index threshold, the method could identify high-grade meningiomas with 90.2% sensitivity and 72.2% specificity. Thus, this quick technique can help surgeons adjust their approach during surgery [50]. Another study by Vartholomatos et al., utilizing the rapid Ioannina Protocol, assessed glioma grades and resection margins. From the 81 patients studied, high-grade gliomas had a notably higher tumor index than low-grade ones. A tumor index threshold was identified that could distinguish between low- and high-grade gliomas with 61.4% sensitivity and 100% specificity. Additionally, iFC confirmed the presence of malignant tissue in all samples from glioma margins, aligning with histology results. In conclusion, iFC offers a promising intraoperative tool for glioma evaluation, though further comparative studies are needed [51].

In the clinical arena, iFC has proven invaluable in surgeries from less malignant meningiomas [60] to aggressive brain tumors, like glioblastomas, where distinguishing tumor margins can be particularly challenging [61]. By offering near-real-time feedback, surgeons can ensure they remove as much of the tumor as possible while minimizing damage to healthy tissue. However, iFC is not devoid of challenges. The technique requires specialized equipment and trained personnel, which might not be readily available in all medical facilities. Furthermore, while iFC is excellent for distinguishing between malignant and nonmalignant cells, it might not always provide detailed insights into the tumor’s histological subtype, which can be crucial for treatment planning [62].

## 5. Comparative Analysis

In the rapidly advancing field of brain malignancy diagnostics and treatment, a plethora of next-generation evaluation techniques, from fast biopsy to iMRI and 5-ALA to intraoperative flow cytometry (iFC), have emerged as potential game-changers. However, when juxtaposed, their unique strengths and limitations become evident, offering insights into their optimal applications. This section offers a comparative analysis, highlighting the unique strengths and potential limitations of each technique.

### 5.1. Speed, Efficiency, and Accuracy

Fast biopsy techniques, as the name suggests, prioritize rapid tissue sampling and analysis. They are especially valuable in scenarios where a quick diagnosis is paramount, such as emergency situations or when immediate surgical intervention is necessary. While fast biopsy techniques prioritize rapid tissue sampling and analysis, iMRI provides real-time imaging feedback during surgeries, and 5-ALA offers immediate visualization of tumor cells through fluorescence. iFC, on the other hand, shines during the surgical procedure itself, offering a near-real-time cellular analysis that can guide the surgeon, ensuring maximal tumor removal and minimal healthy tissue damage.

Each technique, in its own right, enhances the speed and efficiency of diagnosis and treatment. While fast biopsy techniques are designed for speed, there is a potential trade-off in terms of the depth of histopathological analysis. Rapid processing can sometimes compromise slide quality, leading to potential diagnostic inaccuracies. iFC, while providing immediate feedback on cell malignancy, might not always offer detailed insights into the tumor’s histological subtype. Traditional histopathological methods, though slower, still remain the gold standard for a comprehensive tumor characterization. A comparative analysis is presented in Table 1.

A direct analysis of all techniques would require a comparative study in the same clinical setting. Such analyses are only available for 5-ALA and iMRI. 5-ALA’s fluorescence guidance ensures clear delineation between malignant and healthy tissues, though its effectiveness can vary across tumor types. iMRI, on the other hand, offers unparalleled imaging clarity, aiding in precise tumor resections. Because high-grade gliomas (HGGs) have a poor prognosis, patient outcomes largely depend on the extent of tumor removal. A study by Golub et al. compared the effectiveness of 5-ALA and iMRI in optimizing HGG resection. Both iMRI and 5-ALA were found to be superior to conventional neuronavigation in achieving complete tumor removal. However, when comparing iMRI to 5-ALA directly, neither method was definitively more effective [63]. Another study by Naik et al. compared three intraoperative navigation methods: 5-ALA, fluorescein sodium (FS), and iMRI and their combinations to determine which was most effective in maximizing tumor removal. All navigation methods were superior to no navigation, with IMRI + 5-ALA and IMRI alone ranking the highest. FS and IMRI were associated with improved overall and progression-free survival, respectively. Both studies highlighted the superiority of intraoperative navigation methods, specifically iMRI and 5-ALA, in achieving complete resection of high-grade gliomas, with potential benefits in overall and progression-free survival [64]. In conclusion, while both iMRI and 5-ALA are effective, iMRI, especially when combined with 5-ALA, seems to have a slight edge in terms of accuracy for achieving GTR based on the presented studies.

### 5.2. Invasiveness, Patient Impact, and Ease of Use

Fast biopsy techniques, being minimally invasive, generally result in quicker patient recovery and a reduced risk of complications. All three techniques demand specialized equipment and trained personnel. While fast biopsy techniques require expertise in targeted tissue sampling, iMRI necessitates a surgical suite equipped with MRI capabilities and trained staff. 5-ALA, though less resource-intensive, requires expertise in fluorescence-guided surgery. iFC, being an intraoperative technique, does not add additional invasiveness to the surgical procedure but does enhance the surgical process’s precision and efficacy.

Fast biopsy techniques, especially those integrated with real-time imaging, require advanced equipment and trained personnel for optimal results. iFC, being a more specialized technique, demands not only advanced flow cytometry equipment but also personnel trained in both surgical and cytometric procedures. This can limit its availability to more advanced medical facilities with the necessary resources.

All the aforementioned techniques seem to have promising futures. Fast biopsy techniques, with advancements in imaging and molecular biology, might achieve even quicker results with enhanced accuracy. iFC, as technology becomes more accessible and training more widespread, could become a staple in brain tumor surgeries, ensuring better patient outcomes.

## 6. Future Prospects

The realm of brain malignancy diagnostics and treatment is in a state of constant evolution, driven by technological advancements, research breakthroughs, and a deeper understanding of tumor biology. Both fast biopsy techniques and intraoperative flow cytometry (iFC) have already made significant strides in recent years, but the horizon promises even more exciting developments. As imaging modalities continue to improve, we can anticipate even more precise targeting during fast biopsy procedures. Enhanced resolution and real-time molecular imaging might allow clinicians to pinpoint not just the tumor’s location but also its molecular characteristics, facilitating personalized treatment approaches. As the benefits of iFC become more widely recognized, there will likely be a push toward more extensive training programs, making the technique more accessible to surgeons worldwide. This could lead to iFC becoming a standard procedure in brain tumor surgeries, ensuring better outcomes for a larger patient population.

The integration of Artificial Intelligence (AI) and Machine Learning (ML) into diagnostic procedures holds immense potential. For fast biopsy techniques, AI could assist in real-time analysis, further speeding up the diagnostic process. In the context of iFC, machine learning algorithms could be trained to recognize subtle patterns, potentially enhancing the accuracy of real-time tumor margin evaluations. This is already a reality for iMRI and the analysis of gliomas [65,66], with promising results. Similar methodologies are already developing for 5′ALA fluorescence analysis [67]. Such results hold promise for the future of the field.

The challenges posed by brain malignancies necessitate a collaborative approach. We can expect more global research initiatives, bringing together experts from various fields, to drive innovations in both fast biopsy techniques and further integration on methodologies that include 5-ALA, iMRI, and iFC. Such collaborations could accelerate the development of novel techniques, refine existing ones, and ensure that advancements benefit patients worldwide.

## 7. Conclusions

Brain malignancies, with their intricate nature and location, have always posed significant challenges in both diagnosis and treatment over the past few decades.

The emergence of fast biopsy techniques, including 5′ALA, iMRI, and iFC, has marked a transformative shift in the way clinicians approach these tumors. While fast biopsy techniques prioritize rapid and minimally invasive tissue sampling, 5′ALA, iMRI, and iFC offer the invaluable advantage of real-time or near-real-time cellular analysis during surgical procedures. From the rapid diagnostic insights offered by fast biopsy techniques to the real-time surgical precision enabled by intraoperative MRI (iMRI), the fluorescence-guided clarity of 5-ALA, and the accurate, quantitative cancer cell phenotypic analysis by iFC, the arsenal of tools available to clinicians has never been more robust.

This review has critically assessed these techniques, highlighting their respective strengths, limitations, and clinical applications. The comparative analysis underscores the unique value each brings to the table, with fast biopsy techniques excelling in rapid diagnosis and iFC enhancing surgical precision. Each technique, with its unique strengths and potential limitations, plays a pivotal role in the overarching goal of improving patient outcomes. As technology and research continue to evolve, these methods will undoubtedly undergo further refinement, paving the way for even more effective and efficient brain malignancy management. Advancements in imaging, the integration of artificial intelligence, and breakthroughs in equipment and software are set to further revolutionize these techniques. Collaborative global research initiatives will undoubtedly play a pivotal role in driving these innovations, ensuring that the benefits reach patients across the world.

In conclusion, while challenges remain, the advancements in fast biopsy and next-generation evaluation techniques represent significant strides in the search for optimal brain malignancy management. As technology and research continue to evolve, there is hope that these techniques will pave the way for even better patient outcomes in the years to come.

## Figures and Tables

**Table 1 cancers-15-04843-t001:** Comparative analysis of intraoperative techniques in brain malignancies.

Technique	Description	Advantages	Limitations	Clinical Applications
Fast Biopsy	Rapid tissue sampling and analysis.	− Speedy diagnosis − Minimally invasive	− Potential compromise in histopathological depth − Inherent risks of invasive procedures	− Emergency diagnosis − Monitoring tumor progression
Intraoperative Flow Cytometry (iFC)	Near-real-time cellular analysis during surgery.	− Immediate feedback on cell malignancy − Enhances surgical precision	− Requires specialized equipment and training − Might not provide detailed histological subtyping	− Surgeries for for several tumor types − Identifying tumor margins
Intraoperative MRI (iMRI)	Real-time imaging during surgical procedures.	− Dynamic feedback during surgery − Ensures preservation of critical brain structures	− Bulky equipment − High setup and maintenance costs	− Surgeries involving tumors near critical brain structures
5-ALA	Fluorescence-guided surgery using 5-aminolevulinic acid.	− Clear visualization of tumor cells − Enhances tumor resection precision	− Effectiveness varies across tumor types − Potential for false positives	− High-grade glioma surgeries − Delineating tumor margins
iKnife	Real-time tissue analysis during surgery using mass spectrometry.	− Immediate feedback on tissue type − Enhances precision in tumor resection	− Requires specialized equipment and training − Metabolic profiles may vary, needing continuous calibration	− Various types of tumor surgeries − Delineating tumor margins in real-time

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
