# Peer review of "Critical Assessment of Cancer Characterization and Margin Evaluation Techniques in Brain Malignancies: From Fast Biopsy to Intraoperative Flow Cytometry"

_cancers, 2023, doi:10.3390/cancers15194843_

Round 1
Reviewer 1 Report
The manuscript by Liaropoulos et al "Critical assessment of cancer characterization and margin evaluation techniques in brain malignancies: from fast biopsy to intraoperative flow cytometry" provides a review of the literature regarding methodologies for intraoperative cancer characterization with an emphasis on novel advancements and the development of intraoperative flow cytometry methodology.
This is a well-written manuscript and contains valuable and up-to-date information on the subject. The selected methods for discussion, iMRI and 5'ALA, apart from iFC are the most used in the field and i agree with the comparative analysis of those in the text.
However, a major point that needs to be addressed is to add relevant information on a couple other new methods such as intraoperative mass spectometry. It is important to add text on the subject and relevant references.
I would suggest such information to be added as a new paragraph or section in the text as well as new lines presented in Table 1.
Following this addition I would encourage publication of the manuscript.
Author Response
Thank you for the review. We have made the necessary changes in the text highlighted in red. In particular we added:
-a new section on other methodologies
-a new element in the table that discuss intraoperative mass spectrometry
Reviewer 2 Report
This is a really well designed review of cancer characterization and margin evaluation techniques, All the techniques are well and critically presented. There is a very good comparison between the techniques. Future prospects and conclusions are very well structured. Reading this review provides sufficient information for anyone interested in the subject.
Very few minor spelling mistakes need correction.
Author Response
Thank you for the review. We appreciate the feedback. We have proofread the text according to your suggestion.
Round 2
Reviewer 1 Report
The manuscript has been improved and the authors have addressed my comments. I endorse publication